# Exploration via Flow-Based Intrinsic Rewards

## Abstract

Exploration bonuses derived from the novelty of observations in an environment have become a popular approach to motivate exploration for reinforcement learning (RL) agents in the past few years. Recent methods such as curiosity-driven exploration usually estimate the novelty of new observations by the prediction errors of their system dynamics models. In this paper, we introduce the concept of optical flow estimation from the field of computer vision to the RL domain and utilize the errors from optical flow estimation to evaluate the novelty of new observations. We introduce a flow-based intrinsic curiosity module (FICM) capable of learning the motion features and understanding the observations in a more comprehensive and efficient fashion. We evaluate our method and compare it with a number of baselines on several benchmark environments, including *Atari* games, *Super Mario Bros.*, and *ViZDoom*. Our results show that the proposed method is superior to the baselines in certain environments, especially for those featuring sophisticated moving patterns or with high-dimensional observation spaces.

## 1 Introduction

Deep reinforcement learning (DRL) algorithms aim at developing the policy of an agent to maximize its cumulative rewards collected in an environment, and have gained considerable attention in a wide range of application domains, such as game playing (Mnih et al., 2015; Silver et al., 2016) and robot navigation (Zhang et al., 2016). Despite their recent successes, however, one of the key constraints of them is the requirement of dense reward signals. In environments with sparse reward signals, it becomes extremely challenging for an agent to explore and learn a useful policy. Although simple heuristics such as $\epsilon$-greedy (Mnih et al., 2013), entropy regularization (Mnih et al., 2016a), and noisy network (Fortunato, 2018) were proposed, they are still far from satisfactory in such environments.

Researchers in recent years have attempted to deal with the challenge by providing an agent with exploration bonuses (also known as "*intrinsic rewards*") to encourage an agent to explore even when the reward signals from environments are sparse. These bonus rewards are associated with state novelty to incentivize an agent to explore those novel states. A number of novelty measurement strategies have been proposed in the past few years, such as the use of information gain (Houthooft et al., 2016), count-based methods utilizing counting tables (Bellemare et al., 2016; Ostrovski et al., 2017), and prediction-based methods exploiting prediction errors of dynamics models (Stadie et al., 2015; Pathak et al., 2017; Burda et al., 2019a;b). These prediction-based methods differ in the targets of prediction. Pathak et al. (2017); Burda et al. (2019a) introduce a forward dynamics model for predicting the feature representation of the next state based on the current state and the action taken by the agent, which is known as next-frame prediction. Next-frame prediction for complex or rapid-changing observations, however, is rather difficult for forward dynamics models, especially when the prediction is made solely based on the current state and the taken action. It has been widely recognized that performing next-frame prediction typically requires complex feature representations (Kingma & Welling, 2014; Goodfellow et al., 2014; Mirza & Osindero, 2014; Lotter et al., 2017; Xue et al., 2016). On the other hand, Burda et al. (2019b) proposed a self-frame prediction strategy by employing a predictor network to predict the feature representation of the current state from a fixed and randomly initialized target network. Nevertheless, the attempt of self-frame prediction to predict the encoded current state inherently neglects motion features in the observations.

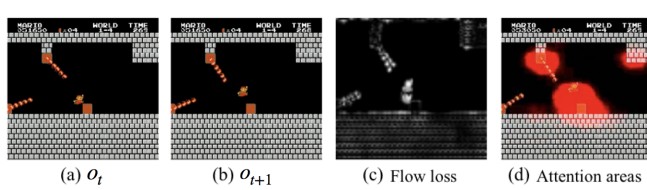

(a) $o_t$     (b) $o_{t+1}$     (c) Flow loss     (d) Attention areas

Figure 1: Illustration of the relation between the prediction errors (i.e., flow loss) and the attention areas of an agent. (a) and (b) are two consecutive frames, (c) visualizes the flow loss, and (d) depicts the attention areas (Greydanus et al., 2018) of an agent.

Figure 2: Prediction errors of FICM versus training iterations for selected states.

Rapid changes or moving patterns in two consecutive observations essentially serve as important signals to motivate an agent for exploration. It is thus inherently reasonable to take these features into account for prediction-based methods. As a result, in this paper we introduce a **f**low-based **i**ntrinsic **c**uriosity **m**odule, called **FICM**, for evaluating the novelty of observations. FICM generates intrinsic rewards based on the prediction errors of optical flow estimation (i.e., the flow loss). Observations with low prediction errors, or low intrinsic rewards, indicate that the agent has seen the observations plenty of times. On the contrary, observations are considered seldom visited when the errors from the predicted flow are non-negligible. The latter case then prompts FICM to generate high intrinsic rewards to encourage the agent for exploration. Fig. 1 provides an illustrative example of FICM for *Super Mario Bros.* It can be observed that the brighter parts of the flow loss in Fig. 1 (c) align with the attention areas (Greydanus et al., 2018) in Fig. 1 (d), implying the significance of motion features in intrinsically motivated exploration. In addition, as exploration bonuses ideally decrease over time, the agent is motivated to continuously explore novel observations during the training phase. Fig. 2 validates that flow loss gradually declines to low values after millions of iterations, indicating the qualification of flow loss to serve as an intrinsic reward signal. In other words, FICM is capable of incentivizing the agent to focus on moving parts of the environment and explore novel observations.

We validate the performance of FICM in a variety of benchmark environments, including *Atari 2600* (Bellemare et al., 2013), *Super Mario Bros.*, and *ViZDoom* (Wydmuch et al., 2018). We demonstrate that FICM is preferable to the previous prediction-based exploration methods in terms of exploration efficiency of the agent in several tasks and environments, especially for those featuring sophisticated moving patterns. We further provide a comprehensive set of ablation analysis for the proposed FICM. The primary contributions of this paper are thus summarized as the following:

- We propose a new flow-based intrinsic curiosity module, called FICM, which leverages on existing methods of optical flow estimation from the field of computer vision (CV) to evaluate the novelty of observations characterizing by sophisticated motion features.

- The proposed FICM encourages an RL agent to learn the motion features and understand the observations from an environment in a more comprehensive manner.

- The proposed FICM is able to encode high dimensional inputs (e.g., RGB frames) and utilizes the information more effectively and efficiently.

- Moreover, the proposed FICM requires only two consecutive frames to obtain sufficient information when estimating the novelty of observations.

The remainder of this paper is organized as the following. Section 2 introduces the background material related to this work. Section 3 presents the proposed FICM framework. Section 4 demonstrates the experimental results and discusses their implications. Section 5 concludes the paper. For more details of the background knowledge, implementations, and setups, please refer to our appendices.

## 2   BACKGROUND

In this section, we provide an overview of the previous curiosity-driven exploration methodologies. We first introduce these methods, followed by a review of the two representative strategies employed by them: the next-frame prediction and the self-frame prediction strategies. To ease the understanding of FICM, we discuss the concepts of DRL as well as optical flow estimation in our appendices.

### 2.1 CURIOSITY-DRIVEN EXPLORATION METHODOLOGIES

Curiosity-driven exploration is an exploration strategy adopted by a number of DRL researchers in recent years (Houthooft et al., 2016; Bellemare et al., 2016; Ostrovski et al., 2017; Pathak et al., 2017; Burda et al., 2019a;b) in order to explore environments more efficiently. While conventional random exploration strategies are easily trapped in local minima of state spaces for complex or spare reward environments, curiosity-based methodologies tend to discover relatively un-visited regions, and therefore are likely to explore more effectively than conventional strategies within the same amount of time. In addition to the extrinsic rewards provided by the environments, most curiosity-driven exploration strategies introduce intrinsic rewards generated by the agent to encourage itself to explore novel states. In this paper, we mainly focus on the prediction-based methods of curiosity-driven exploration including two branches: the next-frame and the self-frame prediction strategies.

#### 2.1.1 NEXT-FRAME PREDICTION STRATEGY

In Pathak et al. (2017); Burda et al. (2019a), the authors formulate the exploration bonuses as the prediction errors of a forward dynamics network, representing an agent's ability to predict the consequence of its actions in a visual feature space. Given an observation $o_t$, a feature representation $\phi_t$ is generated by an embedding network. A forward dynamics network $f$ is then used to predict the next state representation by $\phi_t$ and the agent's taken action $a_t$. The mean-squared error (MSE) $||f(\phi_t, a_t) - \phi_{t+1}||^2$ is minimized by forward dynamics network and serves as the intrinsic reward. After exploring the environment for millions of iterations, more states are visited and the errors decline, indicating the agent are familiar with most of the states. However, next-frame prediction solely based on the current state and the taken action is challenging for complex or rapid-changing observations. Therefore, we propose to employ optical flow estimation to deal with the challenge.

#### 2.1.2 SELF-FRAME PREDICTION STRATEGY

In Burda et al. (2019b), the authors analyze the factors of prediction errors and introduce random network distillation (RND) to measure state novelty. RND is composed of two networks: a fixed and randomly initialized target network $g$, and a predictor network $\hat{g}$. Given an observation $o_t$, the objective of RND is to minimize the MSE $||g(o_t) - \hat{g}(o_t)||^2$. Intuitively, the predictor network can predict correctly on those frequently seen states and produce less intrinsic rewards. By introducing a deterministic target network, RND avoids the problem of stochastic prediction errors in next-frame prediction. However, RND does not consider motion features, which are essential in motivating an agent for exploration. We propose to introduce optical flow estimation to capture those features and encourage our agent to explore an environment from a different perspective, as compared with RND.

## 3 METHODOLOGY

In this section, we present the motivation and design overview of FICM. We first provide an introduction to the concepts of FICM. Then, we formulate these concepts into mathematical equations. Finally, we explore two different implementations of FICM and discuss their features and benefits.

### 3.1 FLOW-BASED CURIOSITY-DRIVEN EXPLORATION

Our objective is to develop a new exploration bonus method that is able to (1) understand the motion features between two consecutive frames, (2) encode the observations efficiently, and (3) estimate the novelty of different observations. As a result, we propose to embrace optical flow estimation (Ilg et al., 2017; Meister et al., 2018), a popular technique commonly used in the field of computer vision (CV) for interpreting displacement of objects in consecutive frames, as our state novelty measurement strategy. FICM is capable of encoding its input observations efficiently even with two consecutive RGB frames (instead of stacked, gray-scale frames), and its efficiency is demonstrated in Section 4.3. Moreover, FICM yields higher intrinsic rewards when the agent encounters unfamiliar states than that in familiar states. It then motivates the agent to revisit those states, and gradually learns the features of them over time, as depicted in Fig. 2. The workflow of the proposed FICM and the architectures of our optical flow predictors are illustrated in Fig. 3 and Fig. 4, respectively.

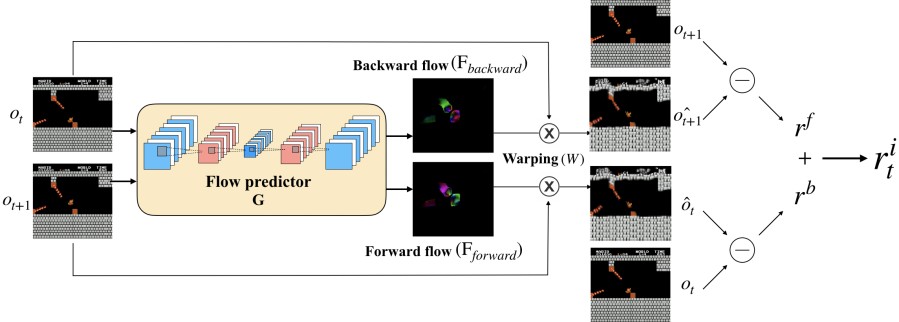

Figure 3: The workflow of the proposed flow-based intrinsic curiosity module (FICM).

## 3.2 FLOW-BASED INTRINSIC CURIOSITY MODULE (FICM)

In this section, we formulate the procedure of FICM as formal mathematical equations. The main objective of FICM is to leverage the optical flow between two consecutive observations as the encoded representation of them. Given two raw input observations $o_t$ and $o_{t+1}$ observed at consecutive timesteps $t$ and $t+1$, FICM takes the 2-tuple $(o_t, o_{t+1})$ as its input, and predicts a forward flow $F_{forward}$ and a backward flow $F_{backward}$ by its flow predictor $G$ parameterized by a set of trainable parameters $\Theta_f$. The two flows $F_{forward}$ and $F_{backward}$ can therefore be expressed as the following:

$$F_{forward} = G(o_t, o_{t+1}, \Theta_f) \quad \text{and} \quad F_{backward} = G(o_{t+1}, o_t, \Theta_f). \tag{1}$$

$F_{forward}$ and $F_{backward}$ are used to generate the observations $\hat{o}_t$ and $\hat{o_{t+1}}$ via a warping function $W(*)$ defined in Fischer et al. (2015); Ilg et al. (2017). The generated $\hat{o}_t$ and $\hat{o_{t+1}}$ are expressed as:

$$\hat{o}_t = W(o_{t+1}, F_{forward}, \beta) \quad \text{and} \quad \hat{o_{t+1}} = W(o_t, F_{backward}, \beta), \tag{2}$$

where $\beta$ is the flow scaling factor. $W(*)$ warps $o_{t+1}$ to $\hat{o}_t$ and $o_t$ to $\hat{o_{t+1}}$ via $F_{forward}$ and $F_{backward}$ respectively using bilinear interpolation and element-wise multiplication with $\beta$. The interested reader is referred to Fischer et al. (2015); Ilg et al. (2017) for more details of the warping algorithm. Please note that in this work, $W(*)$ employs inverse mapping instead of forward mapping to avoid the common duplication problem in flow warping (Beier & Neely, 1992).

With the predicted observations $\hat{o}_t$ and $\hat{o_{t+1}}$, $\Theta_f$ is iteratively updated to minimize the flow loss function $L_G$ of the flow predictor $G$, which consists of a forward flow loss $L^f$ and a backward flow loss $L^b$. The goal of $\Theta_f$ is given by:

$$\min_{\Theta_f} L_G = \min_{\Theta_f} (L^f + L^b) = \min_{\Theta_f} (||o_{t+1} - \hat{o_{t+1}}||^2 + ||o_t - \hat{o}_t||^2), \tag{3}$$

where $(L^f, L^b)$ are derived from the mean-squared error (MSE) between $(o_{t+1}, \hat{o_{t+1}})$ and $(o_t, \hat{o}_t)$, respectively. In this work, $L_G$ is interpreted by FICM as a measure of novelty, and serves as an intrinsic reward signal $r^i$ presented to the agent. The expression of $r^i$ is therefore formulated as:

$$r^i = r^f + r^b = \frac{\zeta}{2}(L^f + L^b) = \frac{\zeta}{2}L_G = \frac{\zeta}{2}(||o_{t+1} - \hat{o_{t+1}}||^2 + ||o_t - \hat{o}_t||^2), \tag{4}$$

where $\zeta$ is the reward scaling factor, and $r^f$ and $r^b$ are the forward and backward intrinsic rewards scaled from $L^f$ and $L^b$, respectively. Please note that $r^i$ is independent of the action taken by the agent, which distinguishes FICM from the intrinsic curiosity module (ICM) proposed in Pathak et al. (2017). FICM only takes two consecutive input observations for estimating the prediction errors of optical flows, which serve as a more meaningful measure for evaluating and memorizing the novelty of observations in environments with high-dimensional observation spaces. The experimental results presented in Section 4 validate the effectiveness of our intrinsic reward $r^i$ and the proposed FICM.

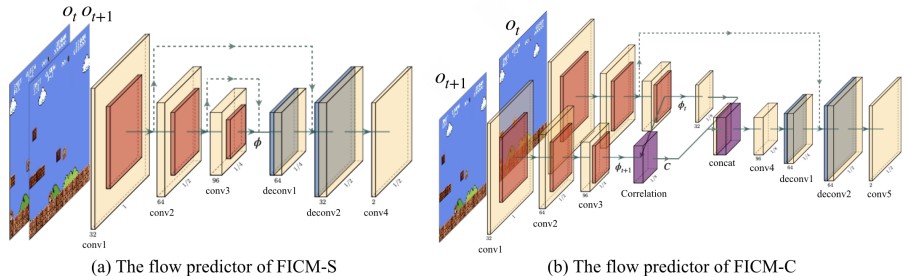

(a) The flow predictor of FICM-S        (b) The flow predictor of FICM-C

Figure 4: The flow predictor architectures of FICM-S and FICM-C.

### 3.3 IMPLEMENTATIONS OF FICM

In this work, we propose two different implementations of FICM: FICM-S and FICM-C. These two implementations adopt different flow predictor architectures based on *FlowNetS* and *FlowNetC* introduced by Ilg et al. (2017), respectively. We employ different implementations to validate that Eqs. (1)-(4) are generalizable to different architectures, rather than restricted to any specific predictor designs. The flow predictor architectures of FICM-S and FICM-C are depicted in Fig. 4 (a) and Fig. 4 (b), respectively. The primary difference between the two predictors lies in their feature extraction strategy from the input observations $o_t$ and $o_{t+1}$. FICM-S encodes the stacked observations $\langle o_t, o_{t+1} \rangle$ together to generate a single feature embedding $\phi$, while the other one encodes $o_t$ and $o_{t+1}$ in different paths to generate two separate feature embeddings $\phi_t$ and $\phi_{t+1}$. In order to preserve both coarse, high-level information and fine, low-level information for enhancing the flow prediction accuracy, the feature embeddings in the two predictor designs are later fused with the feature maps from their shallower parts of the networks by skips (Fischer et al., 2015). The fused feature map is then processed by another convolutional layer at the end to predict the optical flow from $o_t$ to $o_{t+1}$. Please note that the two input paths of Fig. 4 (b) are share-weighted in order to generate comparable embeddings. The flow predictor of FICM-C additionally employs a correlation layer from (Fischer et al., 2015) to perform multiplicative patch comparisons between $\phi_t$ and $\phi_{t+1}$.

## 4 EXPERIMENTAL RESULTS

In this section, we present the experimental results on a number of environments characterizing by complex motion features. We start by comparing the proposed methodology with the previous prediction-based exploration approaches on five *Atari 2600* games (Bellemare et al., 2013) and *Super Mario Bros.*, in the absence of any extrinsic reward signal. Next, we evaluate the performance of FICM on *ViZDoom* (Wydmuch et al., 2018), combining intrinsic rewards with sparse extrinsic rewards. Finally, we present the ablation analysis of FICM from three different aspects. We further provide the training details and the setting of hyper-parameters in the supplementary appendices.

### 4.1 EXPERIMENTS ON PURE EXPLORATION CAPABILITY

**Environments.** We first perform experiments on *Super Mario Bros.* and five different *Atari* games, including *CrazyClimber*, *Enduro*, *KungFuMaster*, *Seaquest*, and *Skiing*. During the training phase, the agents are not provided with any extrinsic reward or end-of-episode signal, in order to examine their abilities of pure exploration.

**Baseline approaches.** We compare the performance of our method against two baselines: (a) next-frame prediction with random CNN features (Burda et al., 2019a), and (b) self-frame prediction (Burda et al., 2019b), denoted as *Forward Dynamics* and *RND*, respectively. Both of the two approaches are prediction-based exploration approaches but differ in their prediction targets. In addition, in order to present the best settings in terms of the agents' performance, our method takes RGB frames as inputs while the baselines remain the default settings described in their papers, that is, taking gray-scale frames as inputs. An ablation study of the usage between RGB and gray-scale frames is further presented in Section 4.3 for fair comparisons.

**Results.** We plot the evaluation curves of our method (denoted as *FICM-C*) and the baselines in different environments in Fig. 5 during the training phase where the agents are trained without access to extrinsic rewards. It is observed that our method significantly outperforms the two baselines in

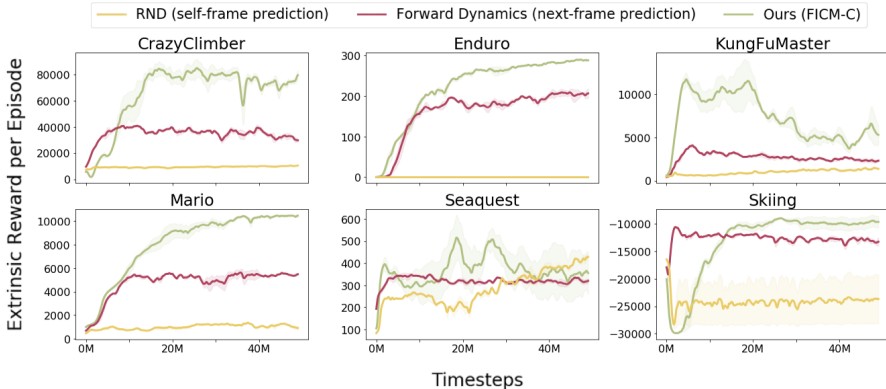

Figure 5: Mean extrinsic rewards (with standard error) that pure exploration agents obtained in *Super Mario Bros.* and five selected *Atari* games. These environments feature complex moving patterns, and therefore our proposed FICM outperforms the baselines in *Super Mario Bros.*, *CrazyClimber*, *Enduro*, *KungFuMaster*, and *Skiing*, while delivering comparable results in *Seaquest*.

games *Super Mario Bros.*, *KungFuMaster*, *CrazyClimber*, *Enduro*, and *Skiing*, while delivering comparable performance to the baselines in games *Seaquest*. These games are characterized by moving objects that require the agents to concentrate on and interact with. As a result, they are favorable to the proposed exploration method, as the optical flow estimator is capable of capturing motion features and perceiving the changes in observations in a more comprehensive fashion.

Since there is no extrinsic reward provided to the agent, the fluctuations and sudden drops in the evaluation curves in Fig. 5 (e.g., *KungFuMaster*) only reflect the exploration processes of the agents. At different timesteps, the agent may focus its attention on exploring different states, leading to fluctuations in the evaluation curve. High scores in the curves indicate that the agents can explore hard-to-visit states (which usually associated with high extrinsic rewards) at certain timesteps. The same fluctuations in the evaluation curves also occurred in the experiments conducted in Burda et al. (2019a). We further illustrate the best extrinsic return curves in our supplementary appendices.

## 4.2 EXPERIMENTS ON EXPLORATION WITH SPARSE EXTRINSIC REWARDS

**Environments.** We further conduct experiments on exploration with sparse extrinsic rewards. The environment we evaluate on is the gaming environment, *DoomMyWayHome-v0* in *ViZDoom* (Wydmuch et al., 2018), the same as those conducted in Pathak et al. (2017). In this environment, the agent is required to reach the fixed goal from certain spawning location in a nine-rooms map, and only receives an extrinsic reward of '+1' if it accomplishes the task. We adopt two setups, *sparse* and *very sparse* reward settings, to evaluate the exploration ability of an agent. The two settings are different in the distance between the initial spawning location of the agent and the goal. The farther the goal is from the spawning location of the agent, the harder the map is for the agent to explore.

**Baseline approaches.** We compare FICM with two baselines, 'ICM + A3C' (denoted as *ICM*) and 'ICM-pixels + A3C' (denoted as *ICM-pixels*) which are both proposed in Pathak et al. (2017). These baselines combine intrinsic curiosity module with training algorithms A3C (Mnih et al., 2016b). *ICM-pixels* is close to *ICM* in architecture except for the absence of the inverse dynamics model, and it computes intrinsic rewards only dependent on forward model loss in next-frame prediction.

**Results.** We analyze and compare the evaluation curves of our proposed methods (denoted as *FICM-C* and *FICM-S*) with those of the baseline methods. The experimental results on the *ViZDoom* environment are depicted in Fig. 6. In the *sparse* reward setting, our methods and the baselines are able to guide the agent to reach the goal. However, in the *very sparse* reward setting, it is observed that *ICM* sometimes suffers from performance drop and is not always able to to obtain the maximum performance, while *ICM-pixels* even fails to reach the goal in this setting. On the contrary, our proposed methods based on FICM are able to converge faster than the baselines and maintain stable performance consistently in both the *sparse* and *very sparse* reward settings over different runs.

The summation of extrinsic and intrinsic rewards are provided to the agents in this *ViZDoom* experiment, with the same setting adopted in Pathak et al. (2017). Although directly adding them up may

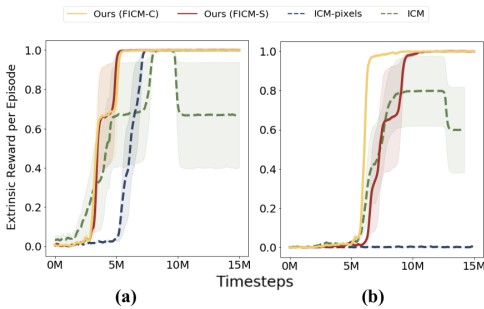 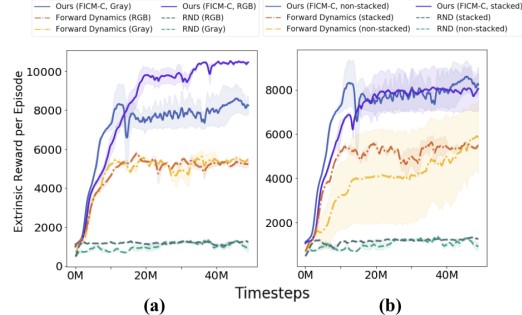

Figure 6: Comparison of the evaluation curves in ViZDoom with (a) sparse and (b) very sparse extrinsic rewards.

Figure 7: Analysis of feature encoding ability of (a) RGB versus gray-scale frames and (b) stacked versus non-stacked frames in *Super Mario Bros.*

be the most straightforward approach, it is definitely not the best way to do so since the objective of extrinsic and intrinsic rewards might be different and even conflicted with each other. Therefore, combining them requires sophisticated engineering. Take *Atari* games for example, the agent may receive rewards from the environment by eliminating an enemy or collecting coins, however, the intrinsic rewards have no relationship with those extrinsic rewards. Careless combinations of them may confuse the agent during exploration, resulting in unsatisfactory performance. Investigating novel methods to effectively combine extrinsic and intrinsic rewards to encourage an agent for exploration is thus a promising future research direction yet it is beyond the scope of this paper.

## 4.3 Ablation Analysis

In this section, we present a set of ablation analysis to investigate the feature encoding abilities of our method and validate the proposed FICM in terms of three different aspects: (1) the dimensionality of the input frames, (2) the number of the stacked frames, and (3) the applicable domains of FICM.

**RGB versus gray-scale frames.**  We hypothesize that RGB frames contain more useful information than gray-scale ones to be utilized in prediction-based approaches. In order to validate this assumption, we eliminate the procedure of RGB to gray-scale conversion on the input frames when generating intrinsic rewards. For a fair comparison, the RL agents still take gray-scale frames as their inputs. We examine whether FICM and the baselines, *Forward Dynamics* and *RND* mentioned in Section 4.1, are able to make use of the extra information to guide the agent for exploration. The experiments are evaluated on *Super Mario Bros.*, and the results are plotted in Fig. 7 (a). We present the results with and without RGB to gray-scale conversion as two different settings for both FICM and the baseline. According to the figure, the baselines perform nearly the same for both settings. In contrast, FICM using RGB frames outperforms the one using only gray-scale frames, indicating that FICM can encode the features more efficiently and utilize the information contained in RGB channels. The superior performance is mainly since the optical flow estimator is more capable of perceiving the motions of objects with color information, as it is much easier to distinguish the difference between foreground and background. As a result, FICM is able to predict optical flow more accurately, and the accurately predicted flow allows FICM to warp the observations more precisely. The increase of input channels does not harm the ability of the flow predictor or warping function, as the same displacement vectors are used in each channel to get the warping results. On the contrary, it is difficult for *Forward Dynamics* and *RND* to obtain appropriate representations of high-dimensional inputs (i.e., RGB frames), since they are lack of eligible embedding networks to encode such RGB information. Fig. 7 (a) reveals their inefficiency in encoding RGB information.

**Stacked versus non-stacked frames.**  We conduct another experiment to investigate the encoding efficiency of FICM and the baselines. We consider two different setups in terms of numbers of stacked frames: (a) two consecutive single frames (denoted as *non-stacked*) and (b) a pair of four stacked frames (denotes as *stacked*). In addition to the existing setups adopted in Section 4.1, we introduce their counterpart setups. The comparison results are plotted in Fig. 7 (b). The results show that both *FICM-C (stacked)* and *FICM-C (non-stacked)* maintain high and consistent performance, indicating that FICM is able to encode the information efficiently even with only two consecutive frames. Furthermore, it is observed that the evaluation curve of *Forward Dynamics (non-stacked)*

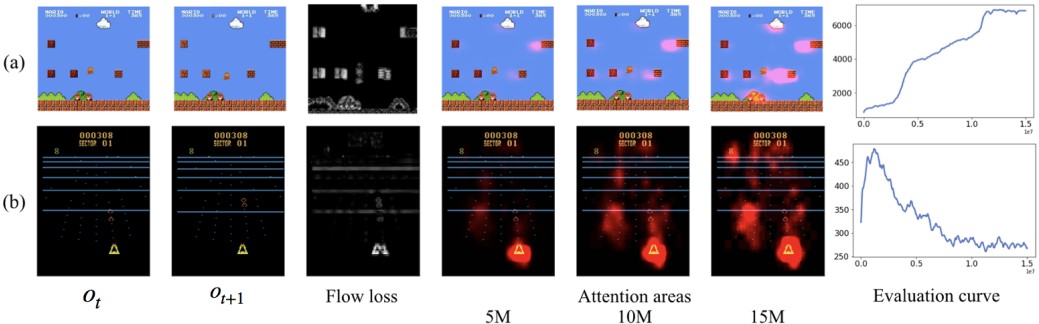

Figure 8: Visualization of flow loss and attention areas in (a) *Super Mario Bros.* and (b) *BeamRider*. The first two columns are two consecutive frames $o_t$ and $o_{t+1}$. The third column shows the flow loss of the two frames, where the brighter parts reflect higher flow loss. The following three columns highlight the attention areas of the agents in red at 5M, 10M, and 15M timesteps. The last column shows the evaluation curves of pure exploration, representing the exploration process of the agents.

exhibits high variance highlighted as the wide shaded area of the evaluation curve, showing that *Forward Dynamics (non-stacked)* suffers from unstable performance due to its poor encoding efficiency. Meanwhile, there is no much difference on performance between *RND (stacked)* and *RND (non-stacked)*. Intuitively, self-frame prediction is less affected by stacked or non-stacked frames. Since there is no significant difference in the evaluation curves of *FICM-C (stacked)* and *FICM-C (non-stacked)*, we suggest that FICM only requires non-stacked frames, to generate intrinsic rewards.

**Discussion of applicable domains.** Despite the outstanding performance in the experiments conducted above, there might be some limitations of FICM which will result in the unsatisfactory results: (1) insufficiency of movements of the major objects (e.g., enemies) and (2) excessive changes of irrelevant components (e.g., backgrounds) that distract the focus of FICM. For instance, if the objects that the agent should pay attention to barely move, FICM generates negligible intrinsic rewards since there is an only little discrepancy between two consecutive frames. On the other hand, if some irrelevant parts of the environments move relatively faster than the crucial objects, FICM may be distracted to concentrate on incorrect regions or components, leading to unsatisfactory performance.

To prove and validate the hypothesis above, we further illustrate how FICM guides the agent to explore the environments by visualizing the flow loss and the attention areas (Greydanus et al., 2018) of the agents during the training phase in Fig. 8. It is observed that the flow loss is highly correlated to the attention areas of the agents at different timesteps for both games. The evidence reveals that FICM is indeed able to guide the agents to concentrate on the regions with high flow loss during the training phase, as the attention areas grow wider and become brighter at those regions in later timesteps. However, flow loss may as well distract the agent to focus on irrelevant parts of the environments. Take *BeamRider* for example, the agent is misled to concentrate on the background by the rich flow loss in it, resulting in its poor performance depicted by the evaluation curve in Fig. 8 (b). On the other hand, the agent is able to master *Super Mario Bros.* as it concentrates on the enemies and is not distracted by irrelevant objects. The visualization further validates that FICM is able to guide an agent to concentrate on moving objects and motivate exploration in an environment.

## 5 CONCLUSIONS

In this paper, we proposed FICM for evaluating the novelty of observations in RL exploration. FICM employs optical flow estimation errors as a measure for generating intrinsic rewards, which allow an RL agent to explore environments featuring moving objects or with high-dimensional observation spaces (i.e., RGB inputs) in a more comprehensive and efficient manner. We demonstrated the proposed methodology and compared it against a number of baselines on *Atari* games, *Super Mario Bros.*, and *ViZDoom*. According to our experiments, we observed that FICM is capable of focusing on essential objects, and guiding the agent to deliver superior performance to the baselines in certain environments. Moreover, we presented a comprehensive set of ablation analysis analyzing the encoding efficiency and applicable domains of FICM, and provided our insights from different aspects. Based on the experimental evidence discussed above, we therefore conclude that FICM is a promising strategy to motivate exploration, especially for complex and rapid-changing observations.

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

## A   APPENDIX

### A.1   BACKGROUND

#### A.1.1   DEEP REINFORCEMENT LEARNING (DRL)

DRL is a method to train an agent to interact with an environment $\mathcal{E}$. At each timesteps t, the agent receives a state $s_t$ form the state space $\mathcal{S}$ of $\mathcal{E}$, takes an action $a_t$ from the action space $\mathcal{A}$ according to its policy $\pi(a|s)$, receives a reward $r(s_t, a_t)$, and transits to next state $s_{t+1} \sim p(s_{t+1}|s_t, a_t)$. The main objective of agent is to maximize its discounted cumulative rewards $R_t = \sum_{i=t}^{T} \gamma^{i-t} r(s_i, a_i)$, where $\gamma \in (0, 1]$ is the discount factor and T is the horizon. The action-value function (i.e., Q-function) of a given policy $\pi$ is defined as the expected return starting from a state-action pair $(s, a)$, expressed as $Q(s, a) = \mathbb{E}[R_t|s_t = s, a_t = a, \pi]$. With the advancement of Deep Neural Networks (DNNs), Deep Q-learning (DQN) (Mnih et al., 2013) was proposed to take advantages of a DNN to approximate the Q-function. Asynchronous Advantage Actor-Critic (A3C) (Mnih et al., 2016b) further introduces asynchronous parallelization and optimization to accelerate the policy and value function updates of an agent. In addition, Proximal Policy Optimization (PPO) (Schulman et al., 2017) takes the biggest possible improvement step on a policy without stepping so far to prevent accidental performance collapse.

#### A.1.2   OPTICAL FLOW ESTIMATION

Optical flow estimation (Fischer et al., 2015) is a technique to evaluate the motion of objects between consecutive images. In usual cases, a reference image and a target image are required. The optical flow is represented as a vector field, where displacement vectors are assigned to certain pixels of the reference image. These vectors represent where those pixels can be found in the target image. In recent years, a number of deep learning approaches running on GPUs dealing with large displacement issues of optical flow estimation have been proposed (Fischer et al., 2015; Schulter et al., 2017; Ilg et al., 2017). Among these techniques, FlowNet 2.0 (Ilg et al., 2017) delivers the most accurate estimation. In this paper, we use simplified network of FlowNet 2.0 to generate optical flow.

### A.2   IMPLEMENTATIONS OF FICM

**FICM-S**   The flow predictor in FICM-S consists of several convolutional and deconvolutional layers. The module first stacks two consecutive observations $o_t$ and $o_{t+1}$ together, and feed the stacked observations $\langle o_t, o_{t+1} \rangle$ into three convolution layers with 32, 64, and 96 filters, respectively, followed by an exponential linear unit (ELU) non-linear activation function. The encoded features are then fused with the feature maps from the shallower parts of the network by adding skips (Fischer et al., 2015; Long et al., 2015), and fed into two deconvolutional layers with 64 and 32 filters. This skip layer fusion architecture allows the flow predictor to preserve both coarse, high layer information and fine, low layer information (Fischer et al., 2015; Long et al., 2015). Finally, the feature map is passed into a convolutional layer with two filters to predict the optical flow from $o_t$ to $o_{t+1}$.

**FICM-C**   The flow predictor in FICM-C encodes two consecutive observations $o_t$ and $o_{t+1}$ separately instead of stacking them together. The input observations are passed through three convolutional layers to generate feature maps $\phi_t$ and $\phi_{t+1}$. The three convolutional layers of the two paths are share-weighted in order to generate better representations of $\phi_t$ and $\phi_{t+1}$, as input observations $o_t$ and $o_{t+1}$ usually contain same or similar patterns. The feature maps $\phi_t$ and $\phi_{t+1}$ are then fed into a correlation layer proposed by Fischer et al. (2015), which performs multiplicative patch comparisons between two feature maps to estimate their correspondences $c$ defined as:

$$c(x_1, x_2) = \sum_{i \in [-k,k] \times [-k,k]} \langle (\phi_t(x_1 + i), \phi_{t+1}(x_2 + i) \rangle, \tag{5}$$

where $x_1$ is the patch center in the first feature map and $x_2$ is that in the second one. Here we fix the maximum displacement $d$ to 2, which means we only compute correlations $c(x_1, x_2)$ limited in a neighborhood of size $D := 2d + 1$ by constraining the range of $x_2$. Since the feature maps of our models have moderate resolutions, $d = 2$ is sufficient to find their correspondences.

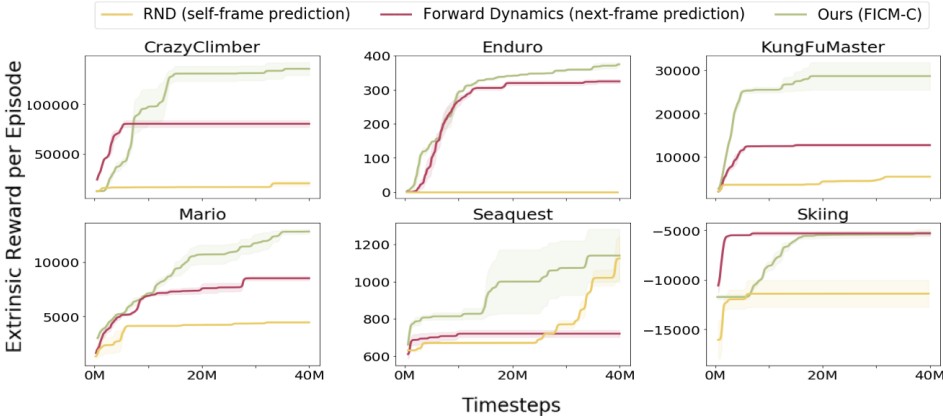

Figure 9: Best extrinsic returns on eight *Atari* games and *Super Mario Bros.*

Once the correspondences $c$ of $\phi_t$ and $\phi_{t+1}$ is estimated, it is concatenated with the feature map from the fourth convolutional layer of the right path in Fig. 4(b). The concatenated feature map is later passed through one convolutional and one deconvolutional layers, fused with the feature map came from the skip path, and then passed through another deconvolutional layer. Similar to Fig. 4(a), the final feature map in Fig. 4(b) is fed into a convolutional layer with two filters to predict the optical flow from $o_t$ to $o_{t+1}$.

### A.3 BEST RETURN IN ATARI GAMES

In addition to mean rewards achieved by the agent, we further illustrate the highest rewards as a measurement of how well the agents explore the environments, since certain states providing higher rewards are harder for the agent to visit. In Fig. 9, it is observed that our method (denoted as *FICM-C*) outperforms the baselines (denoted as *Forward Dynamics*, *RND*) in most of the games, showing consistent results to those in Fig. 5.

### A.4 HYPERPARAMETER

The hyperparameters for the PPO and A3C algorithm are shown in Table 1. We also provide the details of configurations of FICM and baselines in Table 2.

| Hyperparameter | Value |
|---|---|
| **Asynchronous Advantage Actor-Critic (A3C)** | |
| Learning rate of agent | 1e−4 |
| Discount ($\gamma$) | 0.99 |
| GAE parameter ($\lambda$) | 1.0 |
| Entropy coeff. | 0.01 |
| Number of threads | 20 |
| Optimization for RL agent | Adam |
| **Proximal Policy Optimization (PPO)** | |
| Learning rate of agent | 1e−4 |
| Horizon | 128 |
| Discount ($\gamma$) | 0.99 |
| GAE parameter ($\lambda$) | 0.95 |
| Entropy coeff. | 0.001 |
| Number of optimization epochs | 3 |
| Number of minibatches | 8 |
| Number of parallel environments | 128 |
| Optimization for RL agent | Adam |

Table 1: The detail settings of hyper-parameters using in PPO and A3C algorithm.

| Hyperparameter | Value |
|---|---|
| **Experiments on Pure Exploration Capability** | |
| Training algorithm of RL agent | PPO |
| Input size for RL agent | 84*84*4 (gray-scale) |
| Input size for FICM | 84*84*3 (RGB) |
| Learning rate of FICM | $1e-6$ |
| *Forward dynamic* settings | Same as settings provided in (Burda et al., 2019a) |
| *RND* settings | Same as settings provided in (Burda et al., 2019b) |
| **Experiments on Exploration with Sparse Extrinsic Rewards** | |
| Training algorithm of RL agent | A3C |
| Input size for RL agent | 42*42*4 (gray-scale) |
| Learning rate of FICM | $1e-4$ |
| Other baseline settings | Same as settings provided in (Pathak et al., 2017) |
| **Ablation Analysis** | |
| Training algorithm of RL agent | PPO |
| Input size for RL agent | 84*84*4 (gray-scale) |
| Learning rate of FICM | $1e-6$ |
| **RGB versus gray-scale frames** | |
| Input size for FICM | 84*84*3 (RGB) v.s. 84*84*1 (gray-scale) |
| Input size for *Forward dynamic* | 84*84*12 (RGB) v.s. 84*84*4 (gray-scale) |
| Input size for *RND* | 84*84*3 (RGB) v.s. 84*84*1 (gray-scale) |
| **Stacked versus non-stacked frames** | |
| Input size for FICM | 84*84*1 (non-stacked) v.s. 84*84*4 (stacked) |
| Input size for *Forward dynamic* | 84*84*1 (non-stacked) v.s. 84*84*4 (stacked) |
| Input size for *RND* | 84*84*1 (non-stacked) v.s. 84*84*4 (stacked) |
| Other baseline settings | Same as settings provided in (Burda et al., 2019a) and (Burda et al., 2019b) |

Table 2: The detail configurations of FICM and baselines.

