# OpenReview forum: "Exploration via Flow-Based Intrinsic Rewards"
_ICLR.cc/2020/Conference — Reject_

### Official Review · AnonReviewer1 · 2019-10-16
**Official Blind Review #1**

**Rating:** 6

**Review:**



Pros
Solid technical innovation/contribution:
- The paper proposed a novel method FICM that bridged the intrinsic reward in DRL with optical flow loss in CV to encourage exploration in an environment with sparse rewards. To the best of my knowledge, this was the first paper proposed to use moving patterns in two consecutive observations to motivate agent exploration.

Balanced view:
- The authors discussed both the advantages of FICM and settings that FICM might fail to perform well, and conducted experiments to better help the readers understand such nuances. Such balanced view should be valuable to RL communities in both academia and industry.

Clarity:
- In general this was a very well-written paper, I had no difficulty in following the paper throughout. The proposed method (FICM) was clearly motivated, and the authors provided good coverage of related works. Notably, the authors reviewed two relevant methods upon which FICM was motivated, which made the paper self-contained.


Cons
Experiments:
- Experiments were conducted only using a few recent results as baselines (ICM, forward dynamics, RND). It would be interesting to compare FICM against simpler exploration baselines such as epsilon-greedy or entropy regularization.
- I’d also like to see more extensive comparisons between FICM and ICM across different datasets, for example, Super Mario Bros. and the Atari games, instead of only comparing FICM against ICM on ViZDoom.

Significance of the innovation:
- The proposed exploration method seemed to be applicable with a particular RL setting: the environment changes could be represented through consecutive frames (e.g., video games), and optical flow could be used to interpret any object displacements in such consecutive frames. And as the authors discussed, even under such constraints the applicability of proposed method depends on how much changes of the environment were relevant to the goal.

Reproducibility:
- Although the authors discussed the experiment setting in detail in supplements, I believe open-sourcing the code / software used to conduct the experiments would be greatly help with the reproducibility of the proposed method for researchers or practitioners.




Summary
A good paper overall, but the experiments were relatively weak (common for most ICLR submissions) and the novelty was somewhat limited.



**Experience Assessment:**

I have read many papers in this area.

**Review Assessment: Checking Correctness Of Derivations And Theory:**

I assessed the sensibility of the derivations and theory.

**Review Assessment: Checking Correctness Of Experiments:**

I assessed the sensibility of the experiments.

**Review Assessment: Thoroughness In Paper Reading:**

I read the paper thoroughly.

---

> ### Author Response · Authors · 2019-11-07
> **Response to Reviewer #1**
>
> The authors appreciate the reviewer’s time and efforts for reviewing this paper and would like to respond to the questions in the following paragraphs.
>
> [Comment]
> Compare FICM against simpler exploration baselines such as epsilon-greedy or entropy regularization.
> [Response]
> We would like to thank the reviewer for raising this interesting question, and would like to bring to the reviewer's kind attention that in the original paper of our baseline "ICM" [1], the authors had provided a comparison against an ‘A3C’ baseline (using entropy regularization) with epsilon-greedy exploration method (Section 3 of [1]). According to the experimental results presented in Section 4 of [1], it has been demonstrated that ICM is superior to that baseline in a number of environments. This is the reason why we omit that baseline in our paper.  As our primary interest and focus is prediction-based exploration methods using intrinsic reward signals (as discussed in Section 1 of our paper), we only compare our FICM with ICM [1], RND [2] and large-scale [3], concentrating on analyzing the pros and cons between our proposed method and the other prediction-based ones.
>
> However, we would still be glad to include additional comparisons against the suggested methods in the final version of our paper, if the reviewer considers that is informative for the readers to comprehend the paper.
>
> [Comment]
> More extensive comparisons between FICM and ICM across different datasets, for example, Super Mario Bros. and the Atari games, instead of only comparing FICM against ICM on ViZDoom.
> [Response]
> We appreciate the suggestions from the reviewer and would like to share with the reviewer our additional experimental results of ICM using the same hyper-parameter settings described in Section 4.1 in the following figure. (figure link: https://imgur.com/5pPl8PV )
>
> It is observed that ICM is only able to deliver comparable performance to our method in Atari game "Seaquest". We would definitely be glad to incorporate these new results in our manuscript in the revised version.
>
> [Comment]
> Reproducibility.
> [Response]
> Thank you very much for the suggestions.  We have already uploaded our source codes as well as the demonstration videos to the following sites.  Our experimental results and statements presented in the manuscript are fully reproducible and verifiable.
>
> Github: https://github.com/IclrPaperID2276/iclr_paper_2276
> Demo Video: https://youtu.be/JL68QFNj_N8
>
> We hope that we have adequately responded to your questions, and would be very glad to discuss with you if you have any further comments or suggestions.
>
> [1] D. Pathak, P. Agrawal, A. A. Efros, and T. Darrell. Curiosity-driven exploration by self-supervised prediction. In Proc. Int. Conf. Machine Learning (ICML), pp. 2778–2787, May 2017.
> [2] Y. Burda, H. Edwards, A. Storkey, and O. Klimov. Exploration by random network distillation. In Proc. Int. Conf. Learning Representations (ICLR), May 2019b.
> [3] Y. Burda, H. Edwards, D. Pathak, A. J. Storkey, T. Darrell, and A. A. Efros. Large-scale study of curiosity-driven learning. In Proc. Int. Conf. Learning Representation (ICLR), May 2019a.

---

### Official Review · AnonReviewer2 · 2019-10-22
**Official Blind Review #2**

**Rating:** 3

**Review:**

Well motivated paper

The authors study the problem of exploration and exploitation in deep reinforcement learning. The authors propose a new intrinsic curiosity-based method that deploys the methods developed in optical flow. Following this algorithm, the agents utilize the reconstruction error in the optical flow network to come up with intrinsic rewards. The authors show that this approach boosts up the behavior of the RL agents and improves the performance on a set of test environments.

A few comments that I hope might help the authors to improve the clarity of their paper.

1) While the paper is nicely written, I would encourage the authors, of course, if they think necessary, to make the paper slightly more self-contained by explaining the optical flow problem, FlowNet, and warping approach. While a cruise reader might be required to either know literature in optical flow or go and study them along with this paper, it might be helpful for a bit more general readers to have these tools and approaches in access.

2) Regarding the first line of introduction, I would recommend to rephrase it to one imply that the mentioned "aim" is one of the aims of the DRL study.

3) In the fourth line of the intro, the authors mention that the current DRL methods are "constraint" to dense reward. I believe the authors' aim was to imply that these methods perform more desirably in dense reward settings rather than being constrained to such settings.

4) I would also recommend to the authors to elaborate more on the term "attention area" Greydanue et al 2018.

5) It would be helpful to have a better evaluation of this paper if the authors could clarify and motivate the choice of games in their empirical study. For example the empirical study in Fig 5.


6) While I find this study interesting and valuable, the novelty of the approach might fall short to be published at a conference like ICLR with a low acceptance rate. This does not mean that there is anything unscientific about this paper, in fact, the scientific value of this work is appreciated and this work adds a lot to the community.

7) It would also be useful to explicitly explain the advances of this approach over the next frame predictions approaches in stochastic environments. And also, if there is a shortcoming, what are those.

8) Also, what the authors think would happen when the action directly does not change the scene, at least immediately.






**Experience Assessment:**

I have published in this field for several years.

**Review Assessment: Checking Correctness Of Derivations And Theory:**

I carefully checked the derivations and theory.

**Review Assessment: Checking Correctness Of Experiments:**

I assessed the sensibility of the experiments.

**Review Assessment: Thoroughness In Paper Reading:**

I read the paper thoroughly.

---

> ### Author Response · Authors · 2019-11-09
> **Response to Reviewer #2 (part 3/3)**
>
>
> ===Background materials===
> [Optical flow estimation]
> Optical flow estimation is a technique to evaluate the motion of objects between consecutive images. In usual cases, a reference image and a target image are required. The optical flow is represented as a vector field, where displacement vectors are assigned to certain pixels of the reference image. These vectors represent where those pixels can be found in the target image.
>
> In recent years, a number of deep learning approaches running on GPUs dealing with large displacement issues of optical flow estimation have been proposed [1-3]. FlowNet [1] was the pioneer of constructing Convolution Neural Network (CNN) to solve optical flow estimation problem as a supervised task. The author proposed a correlation layer that provides matching capabilities. FlowNet 2.0 [2], an upgraded version of FlowNet, improves the performance in both quality and speed. They adopt a stacked architecture with the auxiliary path to refine intermediate optical flow, and introduce a warping operation which can compensate for some already estimated preliminary motion in the target image. Furthermore, they elaborate on small displacements by introducing a sub-network specializing in small motions. In this paper, we use a simplified version of FlowNet 2.0 to generate optical flow. For more details definition and computation of warping function, we recommend the reviewer can refer to the supplementary materials as provided in [2].
>
> [Attention area]
> The visualization method proposed in [4] is able to visualize the part on which the agent concentrates on current observation. It first selects a region from the original observation and blurs it into a perturbed one. Then, the perturbed observation would be fed to the agent to generate a probability distribution of action to be taken. A score of the importance of the selected region is calculated on the difference between this distribution and the original distribution based on the unperturbed observation. At last, the region with a higher score in observation is colored more brightly.
>
> [1] P. Fischer, A. Dosovitskiy, and E. IlgA. et al. FlowNet: Learning optical flow with convolutional networks. In Proc. IEEE Int. Conf. Computer Vision (ICCV), pp. 2758–2766, May 2015.
> [2] E. Ilg, N. Mayer, T. Saikia, M. Keuper, A. Dosovitskiy, and T. Brox. FlowNet 2.0: Evolution of optical flow estimation with deep networks. In Proc. IEEE Conf. Computer Vision and Pattern Recognition (CVPR), pp. 1647–1655, Dec. 2017.
> [3] Samuel Schulter, Paul Vernaza, Wongun Choi, and Manmohan Krishna Chandraker. Deep network flow for multi-object tracking. Proc. IEEE Conf. Computer Vision and Pattern Recognition (CVPR), pages 2730–2739, Jun. 2017.
> [4] S. Greydanus, A. Koul, J. Dodge, and A. Fern. Visualizing and understanding atari agents. In Int. Conf. Machine Learning (ICML), pp. 1787–1796, Jun. 2018.

---

> > ### Comment · AnonReviewer2 · 2019-11-10
> > **Official Blind Review #2**
> >
> > Dear Authors
> > I appreciate your clear response. I also appreciate your effort in incorporating FICM and the importance of motion in the exploration of RL agents. I will discuss further with other reviewers and the AC, and hopefully, reassess my evaluation accordingly.
> >
> > Cheers,
> > Rev#2
> >
> >
> > (I think it would be in general helpful to incorporate your detailed response in your paper to make it more accessible (it might be quite challenging to do so due to short paper style of conferences). In order to evaluate your paper, I needed to read quite a few other papers in optical flow. It was totally fine to do that, I am not put it as a complaint and the lack of a detailed background did not affect my evaluation of your paper. )

---

> ### Author Response · Authors · 2019-11-09
> **Response to Reviewer #2 (part 2/3)**
>
> [Comment]
> It would also be useful to explicitly explain the advances of this approach over the next frames approaches in stochastic environments. And also, if there is a shortcoming, what are those?
> [Response]
> Thanks for raising this interesting question. We would like to address the reviewer’s concern in two different aspects.
>
> First, we assume that the stochastic environments mentioned by the reviewer correspond to those in which their state transitions are stochastic. In other words, each state transition is associated with a probability, not totally determined by the action performed by the agent. In such a case, we expect that FICM would still be able to learn and generate meaningful intrinsic rewards from the observations, as it does not require the actions performed by the agent for generating intrinsic rewards. What FICM requires, as discussed in Section 3 of our manuscript, are the current observation and the next observation of the agent. As a result, we believe that FICM would demonstrate robustness to stochastic environments. The determining factor of the agent’s performance in such environments would thus greatly rely on the underlying DRL method for learning the policy. On the contrary, ICM would probably not be able to deliver satisfactory performance for such environments. As the state transitions are unpredictable, the intrinsic curiosity modules have no clue to learn the state transition dynamics from the current observation and the action performed by the agent, thereby might cause large prediction errors. Therefore, poor performance caused by the stochastic environment might still be inevitable.
>
> Second, we do have an analysis and discussion regarding the limitations of optical flow. This is why we incorporated additional paragraphs in Section 4.3 for discussing the applicable domains of FICM as a balanced discussion. It is not our paper’s objective to claim or argue that optical flow is omnipotent. Optical flow suffers from occlusions or textureless images, which have already been prevalently recognized by researchers in the domain of computer vision. However, it is still widely adopted in numerous researches as an effective tool for extracting information between consecutive frames. Our research similarly intends to leverage this tool in the domain of reinforcement learning. To validate that the prediction errors from an optical flow estimator can indeed serve as a satisfactory novelty indicator, we presented an experiment in Fig. 2 with a discussion to demonstrate that the prediction errors do gradually decrease over training iterations. This implies that FICM is able to learn and gradually become familiar with the transitions and the motions between consecutive observations.
>
> [Comment]
> What do the authors think would happen when the action directly does not change the scene, at least immediately?
> [Response]
> We would like to thank the reviewer for raising this interesting question. For the scenario mentioned by the reviewer, FICM would generate few intrinsic rewards under such a circumstance, as the transition between the current state and next state is negligible. The agent would therefore be motivated to explore other states. However, if unfamiliar uncontrollable moving objects suddenly appear in the current observation of the agent, FICM would generate intrinsic rewards to encourage the agent to explore the current state more.
>
> [Comment]
> Typos and rephrasing suggestions.
> [Response]
> The authors sincerely appreciate the reviewer’s kindness for pointing out typos and providing constructive rephrasing suggestions (e.g., the “aim” issue). We will definitely revise the manuscript according to the suggestions in our final version.

---

> ### Author Response · Authors · 2019-11-10
> **Response to Reviewer #2 (part 1/3)**
>
> The authors appreciate the reviewer’s time and efforts for reviewing this paper, and would like to respond to the questions in the following paragraphs.
>
> [Comment]
> The authors should elaborate more on optical flow problem, Flownet, warping approach, and the term “attention area”.
> [Response]
> We appreciate the reviewer’s thoughtful feedback. We agree with the reviewer and have prepared additional paragraphs at the end of this response post, including the background materials for optical flow, FlowNet, warping approach, as well as attention area. We would be glad to incorporate those paragraphs into our manuscript, and discuss with you should you have any further comments or suggestions regarding the sufficiency of the background material.
>
> [Comment]
> It would be helpful to have a better evaluation of this paper if the authors could clarify and motivate the choice of games in their empirical study. For example the empirical study in Figure 5.
> [Response]
> We would like to thank the reviewer for raising this question, and are glad to share our perspectives with the reviewer. The selection criteria of our environments is determined by the relevance of motions of the foreground and background components (including the controllable agent and the uncontrollable objects) to the performance (i.e., obtainable scores) of the agent. As the primary theme of this work is to leverage flow features as intrinsic reward signals, we benchmarked our methodology on Atari and Super Mario Bros game environments characterizing sophisticated motions of objects. Taking the Atari game “Enduro” for example. The agent not only has to understand the motion of its controllable car, but is also required to perceive and comprehend the motions of the other cars, as their motions are directly related to the final score of this agent. BeamRider, on the other hand, is not considered as an environment satisfying the above property. According to our experiments, our method does assist the agents to explore better and deliver more satisfactory results in the environments satisfying the above criteria. As a result, instead of focusing on those hard-explored environments, the emphasis of this paper is on bringing to the community the existence and effectiveness of flow-based intrinsic rewards, and motivating researchers with a potential direction in their future endeavors. We have therefore dedicated significant portions of our manuscript to demonstrating and validating that FICM is able to master the environments featuring the above property, and is more effective than other intrinsic motivated approaches when motion features play a vital role in determining the performance of the agents.
>
> Moreover, as the necessity of taking complex motion features into account during the exploration phase of an agent becomes critically important for first-person perspective games, we benchmarked the proposed FICM on ViZDoom, and showed that FICM is naturally more capable of capturing motion features than the baseline methods in Section 4 of our manuscript. As human beings and animals inherently tend to be motivated, attracted, and encouraged by moving objects, we consider that our approach aligns with animal instinct, and believe that our work brings a different perspective to the reinforcement learning community.
>
> Furthermore, in order to provide a balanced analysis of FICM as a complete and comprehensive study, we additionally conducted another set of experiments on “BeamRider” to reveal the limitation of FICM and discussed its applicable domains in Section 4.3. Based on the motivations discussed above, we consider that the flow-based intrinsic reward is worth sharing with the community in ICLR. FICM contributes to the concept of employing flow prediction errors to generate intrinsic rewards, which has never been discussed in the literature before. Rather than finding a panacea for RL exploration, we consider that introducing different perspectives of intrinsic rewards to the existing set of approaches is more likely the correct way to proceed.
>
> We hope that the above discussions have adequately responded to the reviewer’s concerns, and hope that the reviewer can take our perspective into consideration.

---

### Official Review · AnonReviewer3 · 2019-10-22
**Official Blind Review #3**

**Rating:** 3

**Review:**

The paper proposes a novel way to formulate intrinsic reward based on optical flow prediction error. The prediction is done with Flownet-v2 architecture and the training is formulated as self-supervision (instead of the ground-truth-based supervised learning in the original Flownet-v2 paper). The flow predictor takes two frames, predicts forward and backward flows, then warps the first/second frame respectively and compares the warped result with real frame. The comparison error serves as the intrinsic reward signal. The results are demonstrated on 7 environments: SuperMario + 5 Atari games + ViZDoom. On those environments, the proposed method performs better or on-par with ICM and RND baselines.

I am leaning towards rejecting this paper. Two key factors motivate this decision.
First, the motivation for this work is not fully clear: why would the error in flow prediction be a good driving force for curiosity? Optical flow has certain weaknesses, e.g. might not work well for textureless regions because it's hard to find a match. Why would those weaknesses drive the agent to new locations?
Second, the choice of tasks where the largest improvement is shown (i.e. 5 Atari games) seems not well-motivated and rather crafted for the proposed method. Those 5 Atari games are not established hard exploration games.

Detailed arguments for the decision above:
[major concerns]
* Analysis is need on how the method deals with known optical flow problems: occlusion, large displacements, matching ambiguities. Those problems don't fully go away with learning and it is unclear how correlated corresponding errors would be with state novelty.
* "Please note that ri is independent of the action taken by the agent, which distinguishes FICM from the intrinsic curiosity module (ICM) proposed in Pathak et al. (2017)" - but would it then be susceptible to spurious curiosity effects when the agent is drawn to motion of unrelated things? Like leaves trembling in the wind. ICM was proposed to eliminate those effects in the first place, but what is this paper's solution to that problem? Furthermore, the experiments on BeamRider show that this concern is not a theoretical one but quite practical.
* "CrazyClimber, Enduro, KungFuMaster, Seaquest, and Skiing" - none of those Atari environments are known to be hard exploration games (which are normally Gravitar, Montezuma Revenge, Pitfall!, PrivateEye, Solaris, Venture according to Bellemare et al "Unifying count-based exploration and intrinsic motivation"). I understand that every game becomes hard-exploration if the rewards are omitted but then there is a question why those particular games. Moreover, if you omit the rewards the question remains how to select hyperparameters of your method. Was the game reward used for selecting hyperparameters? If not, what is the protocol for their selection? This is a very important question and I hope the authors will address this.
* "These games are characterized by moving objects that require the agents to concentrate on and interact with." - this looks like tailoring the task to suit the method.
* Figure 6 - those results are not great compared to the results of Episodic Curiosity: https://arxiv.org/abs/1810.02274 . Maybe this is because of the basic RL solver (A3C vs PPO) but that brings up another question: why are different solvers used for different tasks in this paper? PPO is normally significantly better than A3C, why not use throughout the whole paper?
[minor concerns]
* Figures are very small and the font in them is not readable. Figure 2 is especially difficult to read because the axes titles are tiny.
* "complex or spare reward" -> sparse
* "However, RND does not consider motion features, which are essential in motivating an agent for exploration." - this is unclear, why are those features essential?
* "We demonstrated the proposed methodology and compared it against a number of baselines on Atari games, Super Mario Bros., and ViZDoom." - please state more clearly that only 5 out of 57 Atari games are considered, here and in the abstract.
* "Best extrinsic returns on eight Atari games and Super Mario Bros." - but only 5 games are shown, where are the other 3?

Suggestions on improving the paper:
1) Better motivating the approach in the paper would help. Why using the flow prediction error as a curiosity signal?
2) Better motivating the choice of the environments and conducting experiments on more environments would be important for evaluating the impact of the paper.

**Experience Assessment:**

I have published one or two papers in this area.

**Review Assessment: Checking Correctness Of Derivations And Theory:**

I assessed the sensibility of the derivations and theory.

**Review Assessment: Checking Correctness Of Experiments:**

I carefully checked the experiments.

**Review Assessment: Thoroughness In Paper Reading:**

I read the paper thoroughly.

---

> ### Author Response · Authors · 2019-11-09
> **Response to Reviewer #3 (part 3/3)**
>
>
> [Comment]
> But would it then be susceptible to spurious curiosity effects when the agent is drawn to motion of unrelated things? Like leaves trembling in the wind. ICM was proposed to eliminate those effects in the first place, but what is this paper’s solution to that problem? Furthermore, the experiments on BeamRider show that this concern is not a theoretical one but quite practical.
> [Response]
> We would like to thank the reviewer for raising the question about “spurious curiosity”. Conventionally, researchers believe that uncontrollable parts (e.g., trembling leaves) in the environment cause spurious curiosity which may mislead an agent’s exploration. Researchers in the past few years have spent tremendous efforts on eliminating such impacts. However, we argue that spurious curiosity is not always caused by uncontrollable parts from an agent’s observations, and not removing them should not be a weakness. In fact, uncontrollable parts sometimes play key roles for effective exploration.
>
> Uncontrollable parts are crucial for success in several games in which other objects’ behaviors are related to the agent’s score. For example, in “Enduro”, comprehending the other cars’ motions is the key to learn a good driving policy. Knowing more about their policies helps the agent make better decisions. However, filtering out uncontrollable parts, as ICM does, prohibits an effective exploration of the others’ acts. This is because the uncontrollable movements of the others might be ignored by ICM. As opposed to ICM, our method preserves the other objects’ motions, enabling effective exploration in games that require the involvement of them. It is worth noticing that in Fig. 5, our method outperforms ICM in “Enduro” by a drastic margin.
>
> On the other hand, uncontrollable parts do hinder the performance of our method in some cases like “BeamRider”. In this game, constantly rolling decorated beams in the game screen are not related to the agent’s scores. Endlessly pursuing curiosity produced by those beams could mislead the exploration direction and thus might result in poor performance. In such a case, filtering out uncontrollable parts could be an answer since focusing on the agent’s motion is the key to success in this game.
>
> To conclude, we believe that removing uncontrollable parts is not a panacea for all scenarios. In fact, whether or not eliminating those uncontrollable is problem-dependent and a tradeoff when designing intrinsic rewards.
>
> [Minor concerns]
> The authors sincerely appreciate the reviewer's kindness for pointing out our typos (e.g., "5 instead of 8" and "sparse") and readability issues, and providing constructive formatting and rephrasing suggestions. We will definitely revise the manuscript according to the suggestions in our final version.
>
> [1] Y. Burda, H. Edwards, D. Pathak, A. J. Storkey, T. Darrell, and A. A. Efros. Large-scale study of curiosity-driven learning. In Proc. Int. Conf. Learning Representation (ICLR), May 2019a.
> [2] D. Pathak, P. Agrawal, A. A. Efros, and T. Darrell. Curiosity-driven exploration by self-supervised prediction. In Proc. Int. Conf. Machine Learning (ICML), pp. 2778–2787, May 2017.

---

> > ### Comment · AnonReviewer3 · 2019-11-11
> > **Reply**
> >
> > I would like to thank the authors for explaining their position on the questions raised in my review.
> >
> > However, my second major concern -- about the choice of environments for evaluating the method -- remains not fully addressed. To address it, I would suggest that the authors evaluate the method on more Atari environments. In particular, I would like to see results on the 6 established hard exploration environments: Gravitar, Montezuma Revenge, Pitfall!, PrivateEye, Solaris, Venture (according to Bellemare et al "Unifying count-based exploration and intrinsic motivation"). It would be great to see how the method performs in no-reward as well as original sparse-reward setting for those environments.
> >
> > "From our perspective, carrying out experiments on tailored environments is not evil. Every method has its own niche. We do believe different types of intrinsic rewards have their best fit for different scenarios, and it is difficult to find one approach being suitable for every situation. As a result, a single and unified algorithm should not be the ultimate goal of research, and is absolutely not our primary purpose and original intention." -- still, demonstrating how a method works on a wider range of environments is the key for understanding the advantages and disadvantages of the method. Even if the method does not perform well, there is a big difference between deteriorating performance slightly or greatly.

---

> > > ### Author Response · Authors · 2019-11-14
> > > **Response to Reviewer #3**
> > >
> > >
> > > The authors appreciate the perspective shared by the reviewer. To address the second concern from the reviewer, we performed further experiments on the suggested six established hard exploration environments with original sparse reward settings, as in [1]. We compared our proposed method with forward dynamics (Random CNN) in [2], which is similar to one of the baselines "Dynamics" in [3].
> > >
> > > The figures of the six environments can be accessed at the following link, https://imgur.com/Jsp3eSm
> > >
> > > These experiments are evaluated for 15M timesteps (~900 parameter updates compared to the experiments in [3]). Please refer to [3] for the results of RND. From the above figure, it can be observed that FICM performs comparably to the forward dynamics baseline in Gravitar, Pitfall, and Solaris, while both of them are not able to acquire any reward in “Montezuma’s Revenge”. However, our results reveal that FICM is able to achieve a score of up to 1000 in Venture within less than 15M timesteps (~900 parameter updates), while RND requires 20K~30K updates to reach the same level of scores. On the contrary, the forward dynamics baseline even fails to receive any reward in this environment.
> > >
> > > We respectfully hope that our results and the above discussions could provide a different perspective for the reviewer to reconsider the evaluation. As we mentioned in our first post, every method has its own niche, while a single and unified algorithm is not our primary purpose and original intention. FICM contributes to the concept of employing flow prediction errors from the field of computer vision to generate intrinsic rewards, which has never been discussed in the literature before. Furthermore, the experimental results presented above are fully reproducible and verifiable. Our source code can be accessed at the following link, https://github.com/IclrPaperID2276/iclr_paper_2276
> > >
> > > We would be glad to discuss further with the reviewer, and are willing to provide additional results should they are necessary. We look forward to hearing from the feedback of the reviewer.
> > >
> > > [1] Bellemare, M., Srinivasan, S., Ostrovski, G., Schaul, T., Saxton, D., and Munos, R. Unifying count-based exploration and intrinsic motivation. In Advances in Neural Information Processing Systems, pp. 1471–1479, 2016.
> > > [2] Y. Burda, H. Edwards, D. Pathak, A. J. Storkey, T. Darrell, and A. A. Efros. Large-scale study of curiosity-driven learning. In Proc. Int. Conf. Learning Representation (ICLR), May 2019a.
> > > [3] Y. Burda, H. Edwards, A. Storkey, and O. Klimov. Exploration by random network distillation. In Proc. Int. Conf. Learning Representations (ICLR), May 2019b.

---

> ### Author Response · Authors · 2019-11-09
> **Response to Reviewer #3 (part 2/3)**
>
>
> [Comment]
> The choice of tasks seems not well-motivated and rather crafted for the proposed methods.
> [Response]
> We understand the reviewer’s concerns. However, we do have different perspectives on this issue and would be glad to discuss our points of view with the reviewer in the following two aspects.
>
> First, the selection criteria of our environments is determined by the relevance of motions of the foreground and background components (including the controllable agent and the uncontrollable objects) to the performance (i.e., obtainable scores) of the agent. Taking the Atari game “Enduro” for example. The agent not only has to understand the motion of its controllable car, but is also required to perceive and comprehend the motions of the other cars, as their motions are directly related to the final score of this agent. BeamRider, on the other hand, is not considered as an environment satisfying the above property. Instead of focusing on those hard-explored environments, the main emphasis of this paper is on bringing to the community the existence and effectiveness of flow-based intrinsic rewards, and motivating researchers with a potential direction in their future endeavors. As a result, we have dedicated significant portions of our manuscript to demonstrating and validating that FICM is able to master the environments featuring the above property, and is more effective than other intrinsic motivated approaches when motion features play a vital role in determining the performance of the agents.
>
> Second, even though we did not present experiments for all Atari games, we do believe that our current experiments sufficiently explain and demonstrate the effectiveness of our method. From our perspective, carrying out experiments on tailored environments is not evil. Every method has its own niche. We do believe different types of intrinsic rewards have their best fit for different scenarios, and it is difficult to find one approach being suitable for every situation. As a result, a single and unified algorithm should not be the ultimate goal of research, and is absolutely not our primary purpose and original intention. For example, although RND delivers superior performance in “Montezuma’s Revenge”, it performs poorly in the experiments presented in Fig. 5 of our paper. On the contrary, our method does assist the agents to explore better and deliver more satisfactory results in the environments satisfying the above criteria. In order to provide a balanced viewpoint of FICM, we further conducted another set of experiments on “BeamRider” to reveal the limitation of FICM and discussed its applicable domains in Section 4.3. Therefore, rather than finding a panacea for RL exploration, we consider that introducing different perspectives of intrinsic rewards to the existing set of approaches is more likely the correct way to proceed.
>
> We hope that the above discussions have adequately responded to the reviewer’s concerns, and hope that the reviewer can take our perspectives into consideration.
>
> ===To respond to the reviewer’s detailed comments===
> [Comment]
> If you omit the rewards the question remains how to select hyperparameters of your method. Was the game reward used for selecting hyperparameters? If not, what is the protocol for their selection?
> [Response]
> We would like to bring to the reviewer’s attention that the game rewards are not used to select hyperparameters for either the agent or the intrinsic module. The hyperparameters of the agents are aligned with those of the baselines in each experiment for fair comparisons. Please note that we did not select the hyperparameters by any specific protocol - we just use the same ones as the baselines. Our hyperparameters are provided in our supplementary material. If you are interested, we have already uploaded our source codes as well as the demonstration videos to the following sites. Our experimental results and statements presented in the manuscript are fully reproducible and verifiable.
>
> Github: https://github.com/IclrPaperID2276/iclr_paper_2276
> Demo Video: https://youtu.be/JL68QFNj_N8
>
> [Comment]
> Why are different solvers used for different tasks in this paper? PPO is normally significantly better than A3C. Why isn’t it used throughout the whole paper?
> [Response]
> We would like to thank the reviewer for raising this question. Since we intended to reproduce the results of [2] and compare with them, we directly executed their officially released open-source codes, where the solver is A3C. We only replaced their intrinsic module by our own method for a fair comparison. The same situation applies to our comparisons with [1], where the solver is PPO.

---

> ### Author Response · Authors · 2019-11-09
> **Response to Reviewer #3 (part 1/3)**
>
> The authors appreciate the thoughtful feedback from the reviewer and would like to respond to the questions in the following paragraphs. Please note that we first address the two major concerns from the reviewer, which also cover our responses to a few questions raised in the detailed comments part provided by the reviewer. Then, we respond to the remaining questions raised in the detailed comments part.
>
> ===To address two major concerns===
> [Comment]
> Better motivating the approach in the paper would help. Why using the flow prediction error as a curiosity signal?
> [Response]
> We appreciate the time and efforts of the reviewer to read through the paper thoroughly. As the reviewer has raised concerns about the motivations of FICM, we would definitely love to share our perspectives with the reviewer and expect a rigorous discussion afterward.
>
> Ｗe believe that rapidly changing parts in two consecutive frames, i.e., motion features extracted by a flow predictor, do usually serve as an important indicator of information in an environment. As depicted in Fig. 1, the motions of Mario and the fire traps contain essential information for the agent to perform well in SuperMario Bros. Biologically, human beings and animals also tend to concentrate on motion features of objects. For instance, animals may not be able to memorize the exact appearance of the objects in their habitats, but do posses the capability to discover whether or not unfamiliar newcomers have intruded into their territories. It is a natural instinct that arouses an animal’s curiosity from motions of unfamiliar feature patterns appearing in its field of view. Our FICM is therefore inspired by the observations mentioned above, and is designed to focus on motion features of objects extracted from two consecutive frames by adopting optical flow estimation for evaluating the novelty of the frames.
>
> We do agree with the reviewer’s concern about the limitations of optical flow. This is why we incorporated additional paragraphs in Section 4.3 for discussing the applicable domains of FICM as a balanced discussion. It is not our paper’s objective to claim or argue that optical flow is omnipotent. Optical flow suffers from occlusions or textureless images, which have already been prevalently recognized by researchers in the domain of computer vision. However, it is still widely adopted in numerous researches as an effective tool for extracting information between consecutive frames. Our research similarly intends to leverage this tool in the domain of reinforcement learning. To validate that the prediction errors from an optical flow estimator can indeed serve as a satisfactory novelty indicator, we presented an experiment in Fig. 2 with a discussion to demonstrate that the prediction errors do gradually decrease over training iterations. This implies that FICM is able to learn and gradually become familiar with the transitions and the motions between consecutive observations in spite of those potential problems.
>
> Based on the motivations discussed above, we consider that the flow-based intrinsic reward is worth sharing with the community in ICLR. FICM contributes to the concept of employing flow prediction errors to generate intrinsic rewards, which has never been discussed in the literature before. The concept is proposed to bring new insights to the research community, and provide a potential direction for future enhancements in the realm of intrinsic reward based exploration.

---

### Decision · Program_Chairs · 2019-12-19

**Decision:**

Reject

**Comment:**

This paper proposes a method for improving exploration by implementing intrinsic rewards based on optical flow prediction error. The approach was evaluated on several Atari games, Super Mario, and VizDoom.

There are several strengths to this work, including the fact that it comes with open source code, and several reviewers agree it’s an interesting approach. R1 thought it was well-written and quite easy to follow. I also commend the authors for being so responsive with comments and for adding the new experiments that were asked for.

The main issue that reviewers pointed out, and which I am also concerned about, is how these particular games were chosen. R3 points out that these 5 Atari games are not known for being hard exploration games. Authors did conduct further experiments on 6 Atari games suggested by the reviewer, but the results didn’t show significant improvement over baselines.

I appreciate the authors’ argument that every method has “its niche”, but the environments chosen must still be properly motivated. I would have preferred to see results on all Atari games, along with detailed and quantitative analysis into why FICM fails on specific tasks. For instance, they state in the rebuttal that “The selection criteria of our environments is determined by the relevance of motions of the foreground and background components (including the controllable agent and the uncontrollable objects) to the performance (i.e., obtainable scores) of the agent.” But it doesn’t seem like this was assessed in any quantitative way.  Without this understanding, it’d be difficult for an outsider to know which tasks are appropriate to use with this approach. I urge the authors to focus on expanding and quantifying the work they depict in Figure 8, which, although it begins to illuminate why FICM works for some games and not others, is still only a qualitative snapshot of 2 games. I still think this is a very interesting approach and look forward to future versions of this paper.